# Adverse outcomes and mortality in users of non-steroidal anti-inflammatory drugs who tested positive for SARS-CoV-2: A Danish nationwide cohort study

**Lars Christian Lund**[1ᵒ], **Kasper Bruun Kristensen**[1ᵒ], **Mette Reilev**[1], **Steffen Christensen**[2], **Reimar Wernich Thomsen**[3], **Christian Fynbo Christiansen**[3], **Henrik Støvring**[1,4], **Nanna Borup Johansen**[5], **Nikolai Constantin Brun**[5], **Jesper Hallas**[1], **Anton Pottegård**[1]*

1 Clinical Pharmacology and Pharmacy, Department of Public Health, University of Southern Denmark, Odense, Denmark, 2 Department of Anaesthesia and Intensive Care Medicine, Aarhus University Hospital, Aarhus, Denmark, 3 Department of Clinical Epidemiology, Aarhus University Hospital, Aarhus, Denmark, 4 Biostatistics, Department of Public Health, Aarhus University, Aarhus, Denmark, 5 Department of Medical Evaluation and Biostatistics, Danish Medicines Agency, Copenhagen, Denmark

ᵒ These authors contributed equally to this work.
* apottegaard@health.sdu.dk

**Data Availability Statement:** Due to data protection regulation, data cannot be shared

## Abstract

### Background

Concerns over the safety of non-steroidal anti-inflammatory drug (NSAID) use during severe acute respiratory syndrome coronavirus 2 (SARS-CoV-2) infection have been raised. We studied whether use of NSAIDs was associated with adverse outcomes and mortality during SARS-CoV-2 infection.

### Methods and findings

We conducted a population-based cohort study using Danish administrative and health registries. We included individuals who tested positive for SARS-CoV-2 during the period 27 February 2020 to 29 April 2020. NSAID users (defined as individuals having filled a prescription for NSAIDs up to 30 days before the SARS-CoV-2 test) were matched to up to 4 non-users on calendar week of the test date and propensity scores based on age, sex, relevant comorbidities, and use of selected prescription drugs. The main outcome was 30-day mortality, and NSAID users were compared to non-users using risk ratios (RRs) and risk differences (RDs). Secondary outcomes included hospitalization, intensive care unit (ICU) admission, mechanical ventilation, and acute renal replacement therapy. A total of 9,236 SARS-CoV-2 PCR-positive individuals were eligible for inclusion. The median age in the study cohort was 50 years, and 58% were female. Of these, 248 (2.7%) had filled a prescription for NSAIDs, and 535 (5.8%) died within 30 days. In the matched analyses, treatment with NSAIDs was not associated with 30-day mortality (RR 1.02, 95% CI 0.57 to 1.82, $p$ = 0.95; RD 0.1%, 95% CI −3.5% to 3.7%, $p$ = 0.95), risk of hospitalization (RR 1.16, 95% CI

directly by the authors. Data is accessible to authorized researchers after application to the Danish Health Data Authority. To apply for data and help with the application process, please see https://sundhedsdatastyrelsen.dk/da/forskerservice/ansog-om-data.

**Funding:** The authors received no specific funding for this work.

**Competing interests:** I have read the journal's policy and the authors of this manuscript have the following competing interests: KBK, NBJ, SC, NB, declare no conflicts of interest. RWT and CFC declare no personal conflicts of interest, yet the Department of Clinical Epidemiology is involved in studies with funding from various companies as research grants to and administered by Aarhus University. None of these studies are related to the current study. HS reports personal fees from Bristol-Myers Squibb, personal fees from Novartis, personal fees from Roche, outside the submitted work. AP and JH report participation in research funded by Alcon, Almirall, Astellas, AstraZeneca, Boehringer-Ingelheim, Novo Nordisk, Servier and LEO Pharma, all with funds paid to the institution where they were employed (no personal fees) and with no relation to the work reported in this paper. LCL reports participation in research projects funded by Menarini Pharmaceutical and LEO Pharma, with funds paid to the institution where he was employed (no personal fees) and with no relation to the work reported in this paper. MR reports participation in research projects funded by LEO Pharma, with funds paid to the institution where she was employed (no personal fees) and with no relation to the work reported in this paper.

**Abbreviations:** COVID-19, coronavirus disease 2019; ICU, intensive care unit; NSAID, non-steroidal anti-inflammatory drug; PS, propensity score; RD, risk difference; RR, risk ratio; SARS-CoV-2, severe acute respiratory syndrome coronavirus 2.

0.87 to 1.53, $p = 0.31$; RD 3.3%, 95% CI −3.4% to 10%, $p = 0.33$), ICU admission (RR 1.04, 95% CI 0.54 to 2.02, $p = 0.90$; RD 0.2%, 95% CI −3.0% to 3.4%, $p = 0.90$), mechanical ventilation (RR 1.14, 95% CI 0.56 to 2.30, $p = 0.72$; RD 0.5%, 95% CI −2.5% to 3.6%, $p = 0.73$), or renal replacement therapy (RR 0.86, 95% CI 0.24 to 3.09, $p = 0.81$; RD −0.2%, 95% CI −2.0% to 1.6%, $p = 0.81$). The main limitations of the study are possible exposure misclassification, as not all individuals who fill an NSAID prescription use the drug continuously, and possible residual confounding by indication, as NSAIDs may generally be prescribed to healthier individuals due to their side effects, but on the other hand may also be prescribed for early symptoms of severe COVID-19.

## Conclusions

Use of NSAIDs was not associated with 30-day mortality, hospitalization, ICU admission, mechanical ventilation, or renal replacement therapy in Danish individuals who tested positive for SARS-CoV-2.

## Trial registration

The European Union electronic Register of Post-Authorisation Studies EUPAS34734

## Author summary

### Why was this study done?

- During the early phases of the pandemic of coronavirus disease 2019 (COVID-19), concerns were raised that ibuprofen, a drug commonly used to treat weak pain and fevers, may lead to a more severe course of coronavirus disease.

- If this risk is verified, it would be important to reduce the use of ibuprofen and ibuprofen-like drugs, commonly referred to as non-steroidal anti-inflammatory drugs (NSAIDs), among patients at risk of COVID-19.

### What did the researchers do and find?

- We identified all Danish residents who tested positive for the infectious agent of COVID-19 and grouped them into users and non-users of NSAIDs.

- The risks of being hospitalized, admitted to the intensive care unit, or dying were compared between the 2 groups.

- Overall, risks for all studied outcomes were similar between users and non-users of ibuprofen and other NSAIDs.

### What do these findings mean?

- NSAIDs do not lead to more severe coronavirus disease according to this study.

## Introduction

As severe acute respiratory syndrome coronavirus 2 (SARS-CoV-2) began to transmit and spread in Europe and the United States, a letter suggesting that ibuprofen could influence the prognosis of coronavirus disease 2019 (COVID-19) through upregulation of angiotensin converting enzyme 2 receptors was circulated widely [1]. Because of this letter and case reports of otherwise healthy young patients with severe COVID-19 who had used NSAIDs in the early stage of disease [2], concerns regarding the safety of NSAID use during the COVID-19 pandemic were widely circulated, including warnings against NSAID use in COVID-19 from the French health minister [2] and the World Health Organization. However, the European Medicines Agency stated that evidence to support warnings against use of NSAIDs in COVID-19 was lacking and stressed the need for further evidence on any effects of NSAIDs on disease prognosis in COVID-19 [3].

The available evidence stems mainly from studies on community-acquired bacterial pneumonia and shows that use of NSAIDs is associated with bacterial complications, specifically empyema and lung abscesses [4–7]. For viral illness, use of NSAIDs was not associated with mortality in intensive care unit (ICU) patients with influenza H1N1 during the 2009 pandemic [8], and a recent study found that use of NSAIDs was not associated with mortality in patients hospitalized for influenza [9]. As use of ibuprofen and other NSAIDs is widespread, data on their safety are urgently needed to guide clinicians and patients on how to use NSAIDs during the COVID-19 pandemic. We therefore examined whether use of NSAIDs prior to infection with SARS-CoV-2 was associated with increased risk of hospitalization, ICU admission, and mortality compared to non-use of NSAIDs.

## Methods

We conducted a Danish nationwide registry-based cohort study investigating the association between NSAID use and 30-day mortality and other adverse outcomes, specified as hospitalization, ICU admission, mechanical ventilation, and acute renal replacement therapy, in all patients who tested positive for SARS-CoV-2. For a graphical representation of the study design, see Fig 1. All analyses followed the publicly registered protocol [10], except for a change in the matching algorithm and a post hoc analysis of test-negative individuals (both detailed below). This study is reported as per the Strengthening the Reporting of Observational Studies in Epidemiology (STROBE) guideline (S1 STROBE Checklist). The institutional data protection board at the University of Southern Denmark and the Danish Health Data Authority approved the research project. According to Danish law, studies based entirely on registry data do not require approval from an ethics review board [11]. All source code used to conduct this study is freely available at https://source.coderefinery.org/lcl/nsaid-covid19.

### Data sources

Data on all Danish residents with a positive test for SARS-CoV-2 were obtained from Danish health and administrative registries as described elsewhere [12]. In brief, identification of the study population was based on prospectively collected data on all Danish residents receiving a polymerase chain reaction (PCR) test for SARS-CoV-2 from the Danish Microbiology Database [13]. Data were linked to the Danish Civil Registration System [14], the Danish National Prescription Registry [15], the Danish National Patient Registry [16], and the Danish Register of Causes of Death [17] by means of the unique personal identifier assigned to all Danish residents at birth or immigration. Data on ICU treatment, mechanical ventilation, and renal replacement therapy from the Danish National Patient Registry were supplemented with daily

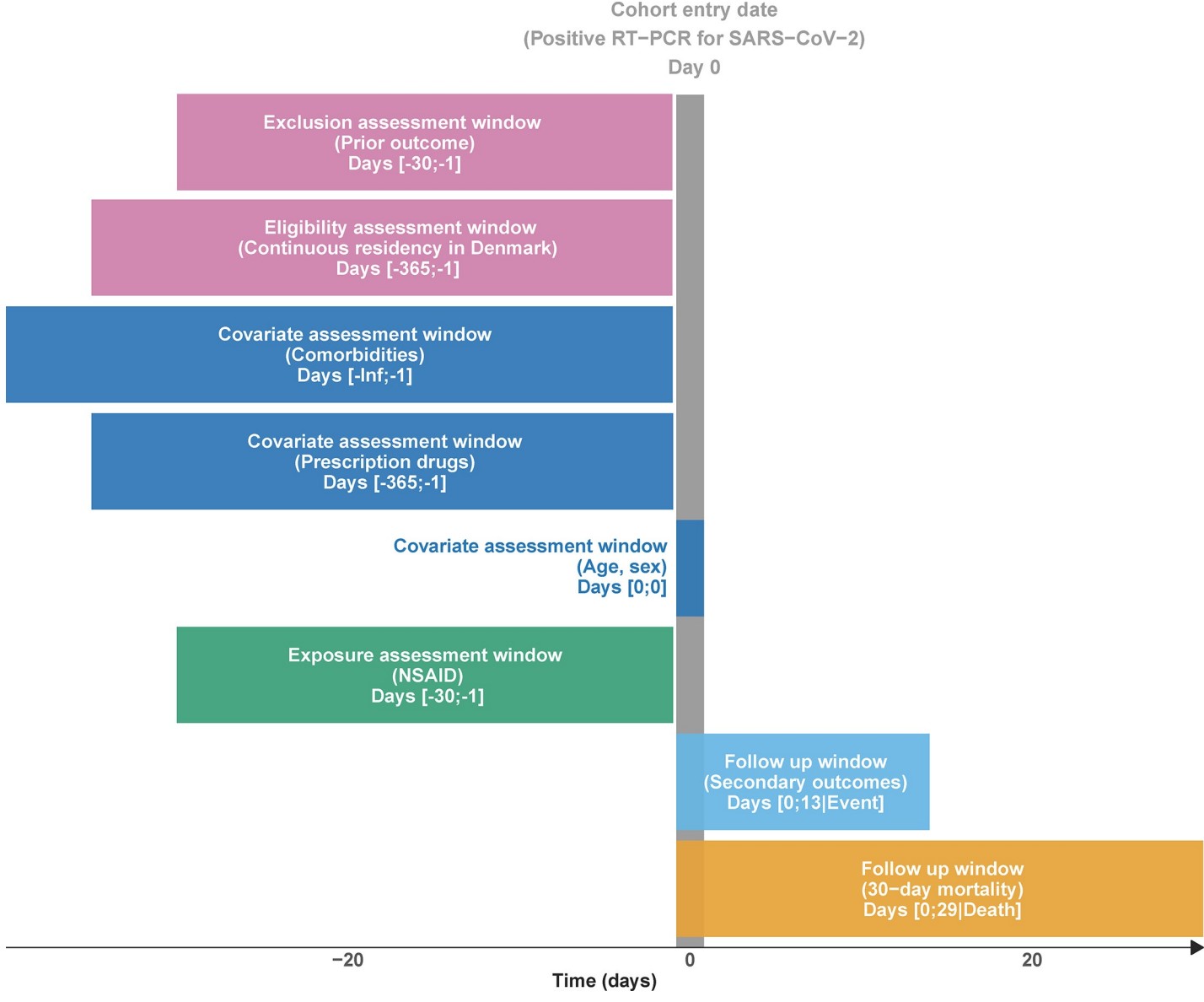

**Fig 1. Study design diagram.** NSAID, non-steroidal anti-inflammatory drug; RT-PCR, reverse transcription polymerase chain reaction; SARS-CoV-2, severe acute respiratory syndrome coronavirus 2.

reports on admitted patients from the 5 Danish regions [16,18]. For more details regarding individual registries, see S1 Appendix.

## Study population

All Danish residents who had a positive PCR test for SARS-CoV-2 during the period 27 February 2020 to 29 April 2020 were included in the study. To ensure complete information on exposure and covariates prior to cohort entry, individuals with less than 1 year of residence in Denmark prior to the positive test for SARS-CoV-2 were excluded. For all secondary outcomes, individuals with an outcome during 30 days to 1 day prior to cohort entry were excluded, partly to ensure that outcomes were incident and plausibly occurring due to

COVID-19 and partly because in-hospital drug use was not available from the Danish National Prescription Registry.

We conducted a post hoc supplementary analysis where we examined the same association in a cohort of all Danish patients who tested negative for SARS-CoV-2 in the study period.

## Exposure

The exposure of interest was current use of any NSAID prior to a positive SARS-CoV-2 test. Current use was defined as having filled a prescription for any NSAID in the 30 days prior to the positive test. Filled NSAID prescriptions were identified from the Danish National Prescription Registry, with information on all dispensed prescriptions at community pharmacies in Denmark since 1995 [19]. Users of NSAIDs were compared to individuals without NSAID use in the corresponding time window. In Denmark, NSAIDs are sold by prescription except for low-dose (200 mg) ibuprofen sold over-the-counter in pack sizes of no more than 20 tablets. In 2018, over-the-counter purchases of ibuprofen constituted 15% of total ibuprofen sales and a smaller proportion of total NSAID sales [20]. Hence, the potential to identify NSAID use from the Danish National Prescription Registry is high compared to many other countries where over-the-counter use of NSAIDs is common [19,21].

## Outcomes

The main outcome of interest was death within 30 days of a positive test for SARS-CoV-2.

The secondary outcomes included hospitalization, ICU admission, mechanical ventilation, and acute renal replacement therapy within 14 days of a positive test for SARS-CoV-2.

## Follow-up

Eligible individuals were followed until the end of follow up (30 days for the main outcome, 14 days for secondary outcomes) or the event of interest.

## Propensity score matching

We used propensity score (PS) matching to increase comparability between NSAID users and non-users. The PS is the estimated probability of receiving the treatment of interest given a set of characteristics [22]. PSs were estimated on the day of the positive SARS-CoV-2 test using logistic regression. Independent variables in the PS model were age, sex, relevant comorbidities, use of selected prescription drugs, and phase of the outbreak. For details of these independent variables, see S2 Appendix. A separate PS was estimated for each exposure definition in the main and supplementary analyses. To evaluate the appropriateness of the model, PS distributions were plotted separately for each cohort and overlap assessed visually. To reduce unmeasured confounding, individuals in the tails of the PS distribution were trimmed asymmetrically [23]. Up to 4 non-users were matched to each NSAID user using a nearest neighbor algorithm. Non-users could be matched to multiple NSAID users, and the maximum allowed difference in the PS between matches was 0.05 [24]. The Danish SARS-CoV-2 test strategy was subject to marked changes from a limited capacity setting in the beginning of the study period to a setting where widespread testing was available at the end of the study period. To account for this [25], we included calendar week of the test date as a forced matching variable. This decision was made post hoc, i.e., not included in the protocol. Covariate balance before and after matching was assessed using standardized mean differences [26].

## Statistical analyses

Descriptive statistics were used to describe NSAID users and non-users at baseline. Continuous variables were reported as medians and interquartile ranges. Dichotomous variables were reported as frequencies and percentages.

Risks and risk differences (RDs) were estimated using generalized linear models with a binomial distribution and an identity link. Risk ratios (RRs) were estimated similarly but using a log link. Matched analyses were implemented using frequency weighting, i.e., NSAID users were assigned a weight of 1, and non-users' weights were assigned according to each individual user's number of matches. Robust 95% confidence intervals were calculated using the sandwich estimator of variance where the assumption regarding independence of observations was relaxed in the matched analyses. Data management and statistical analyses were performed using Stata 16 MP. The codes used to define exposures, covariates, and outcomes are available in S2 Appendix.

## Subgroup analyses

To explore treatment effect heterogeneity, we repeated the main analyses stratifying by age (<65 years, 65+ years), sex, and history of cardiovascular disease. To examine whether widespread testing of healthcare workers influenced the findings, we repeated the main analyses excluding healthcare workers from the study population. We used the same PS as estimated in the main analyses for the subgroup analyses [27].

## Supplementary analyses

We conducted the following supplementary analyses. (1) We relaxed the exposure definition by using an extended NSAID exposure assessment window of 60 days prior to the positive test and repeated the main analyses with this exposure definition. (2) To explore whether reverse causation may have influenced the findings, effect estimates were obtained using an exposure assessment window of 60 days to 14 days before cohort entry (i.e., disregarding NSAID prescriptions filled during the 14 days immediately prior to cohort entry). (3) To evaluate the robustness of the findings with regards to the chosen outcome assessment windows, we obtained 60-day risk estimates for mortality and 30-day risk estimates for secondary outcomes. (4) To examine the potential for residual confounding, we conducted a negative control analysis by repeating the main analyses within the test-negative population, i.e. individuals who tested negative for SARS-CoV-2 (and did not later test positive). If an individual was tested more than once, the first test date was used as the cohort entry date. This post hoc analysis was not specified in the protocol.

## Results

We identified 9,370 individuals who tested positive for SARS-CoV-2 during the study period. Of these, 134 were excluded due to migration within 1 year prior to cohort entry, resulting in an eligible population of 9,236 individuals followed for a total of 705 person-years. The median age in the study cohort was 50 years, and 58% were female. Overall, 535 individuals (5.8%) died within 30 days, 1,512 (16%) were hospitalized within 14 days, 290 (3.1%) were admitted to the ICU, 235 (2.5%) received mechanical ventilation, and 61 (0.7%) received acute renal replacement therapy. In total, 248 (2.7%) patients had filled a prescription for an NSAID within 30 days before the test date. Compared to non-users, NSAID users were older (median age 55 versus 49 years) and more likely to have a history of hospital-diagnosed overweight or obesity (13% versus 9%), to have medical indications for NSAID use such as osteoarthritis

**Table 1. Baseline characteristics in the unmatched and propensity-score-matched cohorts.**

| Characteristic | Unmatched | | | Matched | | |
|---|---|---|---|---|---|---|
| | NSAID users (*n* = 248) | Non-users (*n* = 8,988) | SMD | NSAID users (*n* = 224) | Non-users (*n* = 896) | SMD |
| **Age in years, median (IQR)** | 55 (43–64) | 49 (35–63) | 0.24 | 54 (43–64) | 54 (41–66) | 0.00 |
| **Sex male** | 99 (39.9) | 3,793 (42.2) | 0.05 | 90 (40.2) | 375 (41.9) | 0.03 |
| **Prescription drugs**\* | | | | | | |
| Antihypertensive | 72 (29.0) | 2,221 (24.7) | 0.10 | 62 (27.7) | 233 (26.0) | 0.04 |
| Antidiabetic drug | 26 (10.5) | 680 (7.6) | 0.10 | 21 (9.4) | 78 (8.7) | 0.02 |
| Low-dose aspirin | 16 (6.5) | 532 (5.9) | 0.02 | 15 (6.7) | 47 (5.2) | 0.06 |
| Immunosuppressant | (*n* < 5) | 63 (0.7) | 0.05 | (*n* < 5) | 6 (0.7) | 0.07 |
| Opioid | 59 (23.8) | 950 (10.6) | 0.36 | 46 (20.5) | 172 (19.2) | 0.03 |
| Z-drug | 8 (3.2) | 279 (3.1) | 0.01 | 7 (3.1) | 28 (3.1) | 0.00 |
| Benzodiazepine | 10 (4.0) | 378 (4.2) | 0.01 | 10 (4.5) | 38 (4.2) | 0.01 |
| First generation antipsychotic | (*n* < 5) | 58 (0.6) | 0.03 | (*n* < 5) | (*n* < 5) | 0.02 |
| Second generation antipsychotic | (*n* < 5) | 224 (2.5) | 0.10 | (*n* < 5) | 11 (1.2) | 0.03 |
| Systemic glucocorticoid | 19 (7.7) | 431 (4.8) | 0.12 | 15 (6.7) | 65 (7.3) | 0.02 |
| Inhaled corticosteroid | 27 (10.9) | 625 (7.0) | 0.14 | 21 (9.4) | 92 (10.3) | 0.03 |
| **Prior diagnoses**\*\* | | | | | | |
| Asthma | 16 (6.5) | 613 (6.8) | 0.01 | 13 (5.8) | 47 (5.2) | 0.02 |
| COPD | 11 (4.4) | 368 (4.1) | 0.02 | 9 (4.0) | 35 (3.9) | 0.01 |
| Cardiovascular disease | 28 (11.3) | 1,238 (13.8) | 0.08 | 23 (10.3) | 91 (10.2) | 0.00 |
| Ischemic stroke | 9 (3.6) | 376 (4.2) | 0.03 | 8 (3.6) | 30 (3.3) | 0.01 |
| Chronic kidney failure | (*n* < 5) | 126 (1.4) | 0.11 | (*n* < 5) | (*n* < 5) | 0.06 |
| Liver disease | (*n* < 5) | 125 (1.4) | 0.02 | (*n* < 5) | 10 (1.1) | 0.06 |
| Alcohol-related disorders | 5 (2.0) | 239 (2.7) | 0.04 | (*n* < 5) | 12 (1.3) | 0.04 |
| Dementia | (*n* < 5) | 154 (1.7) | 0.08 | (*n* < 5) | 10 (1.1) | 0.02 |
| Cancer | 21 (8.5) | 646 (7.2) | 0.05 | 16 (7.1) | 64 (7.1) | 0.00 |
| Overweight or obesity | 33 (13.3) | 765 (8.5) | 0.15 | 29 (12.9) | 111 (12.4) | 0.02 |
| Hemiplegia and paraplegia | (*n* < 5) | 35 (0.4) | 0.00 | (*n* < 5) | (*n* < 5) | 0.02 |
| Osteoarthritis | 47 (19.0) | 1,054 (11.7) | 0.20 | 37 (16.5) | 143 (16.0) | 0.02 |
| Rheumatoid arthritis | 17 (6.9) | 308 (3.4) | 0.16 | 13 (5.8) | 51 (5.7) | 0.00 |
| Dysmenorrhea | 7 (2.8) | 62 (0.7) | 0.16 | (*n* < 5) | 8 (0.9) | 0.00 |

Data are given as number (percent) unless otherwise indicated.

\*Defined as 1 or more prescription fills during the period 365 days to 1 day prior to cohort entry.

\*\*Defined as 1 or more discharge diagnoses assigned up to 1 day prior to cohort entry.

COPD, chronic obstructive pulmonary disease; IQR, interquartile range; NSAID, non-steroidal anti-inflammatory drug; SMD, standardized mean difference.

(19% versus 12%) or rheumatoid arthritis (7% versus 3%), and to have been prescribed opioids the year before sampling date (24% versus 11%). After matching, covariates were well balanced, with standardized mean differences ≤ 0.1 (Table 1). Use of opioids was strongly associated with use of NSAIDs and 30-day mortality, while cardiovascular disease and dementia was negatively associated with use of NSAIDs and positively associated with death (S1 Table).

## Main outcomes

NSAID use was not associated with 30-day mortality in the crude (unmatched) analyses or adjusted (matched) analyses (Table 2). In the adjusted analyses, the 30-day mortality rate was 6.3% (95% CI 3.1% to 9.4%) in NSAID users and 6.1% (95% CI 4.4% to 7.8%) in non-users,

**Table 2. Association between current NSAID use and 30-day mortality, hospitalization, ICU admission, mechanical ventilation, and renal replacement therapy in unmatched and propensity-score-matched cohorts.**

| Outcome | NSAID users | | Non-users | | Comparison | | | |
|---|---|---|---|---|---|---|---|---|
| | Number of events/ sample size | Risk (%) (95% CI) | Number of events/ sample size | Risk (%) (95% CI) | Risk difference (%) (95% CI) | p-Value | Risk ratio (95% CI) | p-Value |
| **Unmatched cohort** | | | | | | | | |
| Death | 14/248 | 5.6 (2.8, 8.5) | 521/8,988 | 5.8 (5.3, 6.3) | −0.2 (−3.1, 2.8) | 0.92 | 0.97 (0.58, 1.63) | 0.92 |
| Hospitalization* | 56/228 | 24.6 (19.0, 30.2) | 1,456/8,414 | 17.3 (16.5, 18.1) | 7.3 (1.6, 12.9) | 0.01 | 1.42 (1.13, 1.79) | <0.01 |
| ICU admission* | 11/247 | 4.5 (1.9, 7.0) | 279/8,956 | 3.1 (2.8, 3.5) | 1.3 (−1.3, 3.9) | 0.31 | 1.43 (0.79, 2.58) | 0.23 |
| Mechanical ventilation* | 10/248 | 4.0 (1.6, 6.5) | 225/8,970 | 2.5 (2.2, 2.8) | 1.5 (−0.9, 4.0) | 0.23 | 1.61 (0.86, 2.99) | 0.13 |
| Renal replacement therapy* | $n < 5$/248 | —** | —** | —** | 0.6 (−0.8, 1.9) | 0.42 | 1.87 (0.59, 5.94) | 0.29 |
| **Matched cohort** | | | | | | | | |
| Death | 14/224 | 6.3 (3.1, 9.4) | 55/896 | 6.1 (4.4, 7.8) | 0.1 (−3.5, 3.7) | 0.95 | 1.02 (0.57, 1.82) | 0.95 |
| Hospitalization* | 50/204 | 24.5 (18.6, 30.4) | 175/826 | 21.2 (18.1, 24.3) | 3.3 (−3.4, 10.0) | 0.33 | 1.16 (0.87, 1.53) | 0.31 |
| ICU admission* | 11/223 | 4.9 (2.1, 7.8) | 42/889 | 4.7 (3.2, 6.2) | 0.2 (−3.0, 3.4) | 0.90 | 1.04 (0.54, 2.02) | 0.90 |
| Mechanical ventilation* | 10/224 | 4.5 (1.8, 7.2) | 35/891 | 3.9 (2.5, 5.3) | 0.5 (−2.5, 3.6) | 0.73 | 1.14 (0.56, 2.30) | 0.72 |
| Renal replacement therapy* | $n < 5$/224 | —** | —** | —** | −0.2 (−2.0, 1.6) | 0.81 | 0.86 (0.24, 3.09) | 0.81 |

NSAID use was defined as having an NSAID prescription filled within 30 days prior to the date of cohort entry.

*Patients with a secondary outcome occurring during the exclusion assessment window were excluded, resulting in exclusion of $n = 594$ patients for hospitalization, $n = 33$ for ICU admission, $n = 18$ for mechanical ventilation, and $n = 6$ for renal replacement therapy in unmatched cohorts, and $n = 90$, $n = 8$, $n = 5$, and $n < 5$, respectively, in matched cohorts.

**Censored to preserve anonymity for counts $n < 5$.

ICU, intensive care unit; NSAID, non-steroidal anti-inflammatory drug.

corresponding to a RD of 0.1% (95% CI −3.5 to 3.7, $p = 0.95$) and a RR of 1.02 (95% CI 0.57 to 1.82, $p = 0.95$).

## Secondary outcomes

In the crude analyses, use of NSAIDs was associated with an increased risk of hospitalization (RR 1.42, 95% CI 1.13 to 1.79, $p < 0.01$) but was not associated with an increased risk of ICU admission (RR 1.43, 95% CI 0.79 to 2.58, $p = 0.23$), mechanical ventilation (RR 1.61, 95% CI 0.86 to 2.99, $p = 0.13$), or renal replacement therapy (RR 1.87, 95% CI 0.59 to 5.94, $p = 0.29$) (Table 2). After adjustment, NSAID use was not associated with hospitalization (RR 1.16, 95% CI 0.87 to 1.53, $p = 0.31$), ICU admission (RR 1.04, 95% CI 0.54 to 2.02, $p = 0.90$), mechanical ventilation (RR 1.14, 95% CI 0.56 to 2.30, $p = 0.72$), or renal replacement therapy (RR 0.86, 95% CI 0.24 to 3.09, $p = 0.81$).

## Subgroup analyses

The subgroup analyses were limited by low power and wide confidence intervals (Table 3). In individuals below 65 years of age, the adjusted RR for death associated with NSAID use was

**Table 3. Association between current NSAID use and 30-day mortality, hospitalization, ICU admission, mechanical ventilation, and renal replacement therapy in propensity-score-matched cohorts according to subgroups of interest.**

| Subgroup | Outcome | Risk (%) (95% CI) | | Comparison | | | |
|---|---|---|---|---|---|---|---|
| | | NSAID users | Non-users | Risk difference (%) (95% CI) | *p*-Value | Risk ratio (95% CI) | *p*-Value |
| Age < 65 years | Death | 1.2 (−0.4, 2.8) | 0.3 (−0.1, 0.7) | 0.8 (−0.8, 2.5) | 0.31 | 3.76 (0.53, 26.56) | 0.18 |
| | Hospitalization | 16.3 (10.6, 21.9) | 11.2 (8.5, 13.9) | 5.1 (−1.2, 11.3) | 0.11 | 1.45 (0.95, 2.22) | 0.08 |
| | ICU admission | 1.2 (−0.4, 2.8) | 2.5 (1.1, 3.8) | −1.3 (−3.4, 0.8) | 0.22 | 0.47 (0.11, 2.07) | 0.43 |
| | Mechanical ventilation | 1.2 (−0.4, 2.8) | 1.8 (0.7, 3.0) | −0.7 (−2.7, 1.3) | 0.50 | 0.63 (0.14, 2.87) | 0.55 |
| | Renal replacement therapy | 0.6 (−0.6, 1.7) | 1.1 (0.1, 2.1) | −0.5 (−2.0, 1.0) | 0.52 | 0.54 (0.06, 4.67) | 0.57 |
| Age 65+ years | Death | 23.5 (11.8, 35.3) | 21.6 (16.0, 27.3) | 1.9 (−11.1, 14.9) | 0.77 | 1.09 (0.62, 1.91) | 0.77 |
| | Hospitalization | 60.5 (44.8, 76.3) | 54.2 (46.0, 62.3) | 6.4 (−11.2, 23.9) | 0.48 | 1.12 (0.83, 1.51) | 0.47 |
| | ICU admission | 18.0 (7.2, 28.8) | 10.9 (6.6, 15.2) | 7.1 (−4.4, 18.6) | 0.22 | 1.65 (0.81, 3.37) | 0.17 |
| | Mechanical ventilation | 15.7 (5.6, 25.8) | 9.5 (5.5, 13.6) | 6.1 (−4.6, 16.9) | 0.26 | 1.64 (0.76, 3.53) | 0.20 |
| | Renal replacement therapy | 3.9 (−1.5, 9.3) | 2.9 (0.5, 5.2) | 1.1 (−4.8, 6.9) | 0.72 | 1.37 (0.28, 6.73) | 0.70 |
| Female | Death | 4.5 (1.0, 8.0) | 4.8 (2.8, 6.8) | −0.3 (−4.4, 3.7) | 0.88 | 0.93 (0.38, 2.27) | 0.88 |
| | Hospitalization | 21.7 (14.6, 28.8) | 14.0 (10.5, 17.4) | 7.7 (−0.2, 15.6) | 0.05 | 1.55 (1.03, 2.34) | 0.03 |
| | ICU admission | 1.5 (−0.6, 3.6) | 1.3 (0.4, 2.3) | 0.1 (−2.1, 2.4) | 0.90 | 1.11 (0.23, 5.29) | 0.90 |
| | Mechanical ventilation | 1.5 (−0.6, 3.6) | 0.6 (−0.1, 1.2) | 0.9 (−1.2, 3.1) | 0.41 | 2.59 (0.44, 15.36) | 0.30 |
| | Renal replacement therapy | — | 0.8 (0.0, 1.5) | — | — | — | — |
| Male | Death | 8.9 (3.0, 14.8) | 8.0 (5.1, 10.9) | 0.9 (−5.7, 7.5) | 0.79 | 1.11 (0.52, 2.37) | 0.79 |
| | Hospitalization | 29.3 (19.0, 39.7) | 31.9 (26.3, 37.5) | −2.6 (−14.3, 9.2) | 0.67 | 0.92 (0.62, 1.36) | 0.67 |
| | ICU admission | 10.1 (3.8, 16.4) | 9.5 (6.2, 12.8) | 0.6 (−6.5, 7.7) | 0.86 | 1.07 (0.52, 2.17) | 0.86 |
| | Mechanical ventilation | 8.9 (3.0, 14.8) | 8.6 (5.4, 11.8) | 0.3 (−6.4, 7.0) | 0.94 | 1.03 (0.48, 2.20) | 0.94 |
| | Renal replacement therapy | 3.3 (−0.4, 7.1) | 2.7 (0.6, 4.7) | 0.7 (−3.6, 4.9) | 0.76 | 1.25 (0.32, 4.83) | 0.75 |
| No history of cardiovascular disease | Death | 4.5 (1.6, 7.3) | 3.5 (2.2, 4.8) | 1.0 (−2.1, 4.1) | 0.53 | 1.29 (0.62, 2.69) | 0.50 |
| | Hospitalization | 23.9 (17.8, 30.1) | 19.3 (16.2, 22.5) | 4.6 (−2.3, 11.5) | 0.19 | 1.24 (0.91, 1.68) | 0.17 |
| | ICU admission | 4.5 (1.6, 7.3) | 4.1 (2.6, 5.7) | 0.4 (−2.9, 3.6) | 0.83 | 1.09 (0.52, 2.28) | 0.83 |
| | Mechanical ventilation | 4.0 (1.3, 6.7) | 3.5 (2.1, 4.9) | 0.5 (−2.6, 3.6) | 0.75 | 1.14 (0.51, 2.52) | 0.75 |
| | Renal replacement therapy | 1.0 (−0.4, 2.4) | 1.2 (0.3, 2.2) | −0.2 (−1.9, 1.4) | 0.77 | 0.80 (0.16, 3.90) | 0.78 |
| History of cardiovascular disease | Death | 21.7 (4.5, 39.0) | 29.7 (19.0, 40.4) | −7.9 (−28.0, 12.1) | 0.44 | 0.73 (0.31, 1.73) | 0.48 |
| | Hospitalization | 31.3 (7.8, 54.7) | 42.4 (29.2, 55.7) | −11.2 (−37.6, 15.2) | 0.41 | 0.74 (0.33, 1.63) | 0.45 |
| | ICU admission | 9.1 (−3.2, 21.4) | 10.2 (3.8, 16.7) | −1.1 (−14.8, 12.5) | 0.87 | 0.89 (0.20, 3.86) | 0.88 |
| | Mechanical ventilation | 8.7 (−3.1, 20.5) | 7.9 (2.2, 13.5) | 0.8 (−12.1, 13.7) | 0.90 | 1.11 (0.24, 5.02) | 0.90 |
| | Renal replacement therapy | 4.3 (−4.2, 12.9) | 4.4 (0.1, 8.7) | −0.1 (−9.5, 9.3) | 0.98 | 0.98 (0.11, 8.44) | 0.98 |

(*Continued*)

**Table 3.**  (Continued)

| Subgroup | Outcome | Risk (%) (95% CI) | | Comparison | | | |
|---|---|---|---|---|---|---|---|
| | | NSAID users | Non-users | Risk difference (%) (95% CI) | *p*-Value | Risk ratio (95% CI) | *p*-Value |
| Not healthcare professional | Death | 7.0 (3.3, 10.7) | 8.0 (5.7, 10.2) | −0.9 (−5.3, 3.4) | 0.67 | 0.88 (0.49, 1.60) | 0.68 |
| | Hospitalization | 27.9 (21.0, 34.7) | 26.4 (22.5, 30.4) | 1.4 (−6.5, 9.3) | 0.72 | 1.05 (0.79, 1.41) | 0.72 |
| | ICU admission | 5.4 (2.2, 8.7) | 6.1 (4.1, 8.1) | −0.7 (−4.5, 3.1) | 0.73 | 0.89 (0.45, 1.76) | 0.73 |
| | Mechanical ventilation | 4.9 (1.8, 8.0) | 5.1 (3.2, 6.9) | −0.2 (−3.8, 3.4) | 0.92 | 0.96 (0.46, 2.00) | 0.92 |
| | Renal replacement therapy | 1.1 (−0.4, 2.6) | 2.1 (0.8, 3.4) | −1.0 (−3.0, 1.0) | 0.32 | 0.52 (0.12, 2.37) | 0.40 |

NSAID use was defined as a filled prescription within 30 days prior to the date of cohort entry.

ICU, intensive care unit; NSAID, non-steroidal anti-inflammatory drug.

3.76 (95% CI 0.53 to 26.6, *p* = 0.18). After adjustment, use of NSAIDs was associated with hospitalization in females (RR 1.55, 95% CI 1.03–2.34, *p* = 0.03) but not in males (RR 0.92, 95% CI 0.62 to 1.36, *p* = 0.67). Otherwise, the RRs were not modified by patient characteristics.

## Supplementary analyses with different outcome or exposure assessment windows

Increasing the duration of follow-up to 60 days for death and 30 days for hospitalization, ICU admission, mechanical ventilation, and acute renal replacement therapy did not influence the findings (S2 Table).

When extending the definition of current NSAID use to a filled prescription up to 60 days before the test date, we observed similar null findings (S3 Table). Likewise, defining exposure as a filled prescription 60 days to 14 days before the test date yielded comparable results (S4 Table).

## Test-negative individuals

We identified 204,920 individuals with a negative SARS-CoV-2 PCR test in the study period. Of these, 3,506 were excluded due to migration within 1 year prior to cohort entry, resulting in a population of 201,414 individuals followed up for a total of 15,840 person-years. Use of NSAIDs was associated with a decreased risk of death (RR 0.64, 95% CI 0.49 to 0.84, *p* < 0.01) and increased risk of hospitalization (RR 1.18, 95% CI 1.08 to 1.28, *p* < 0.001) and ICU admission (RR 1.39, 95% CI 1.00–1.95, *p* = 0.05) in the adjusted analyses (Table 4).

## Discussion

We examined whether use of NSAIDs was associated with 30-day mortality and adverse outcomes in a nationwide population of SARS-CoV-2-positive individuals. Use of NSAIDs was not associated with increased 30-day mortality, a finding that was robust in a range of supplementary analyses. Likewise, use of NSAIDs was not associated with an increased risk of hospitalization, ICU admission, mechanical ventilation, or renal replacement therapy in the adjusted analyses.

**Table 4. Association between current NSAID use and 30-day mortality, hospitalization, ICU admission, mechanical ventilation, and renal replacement therapy in unmatched and propensity-score-matched cohorts of individuals who tested negative for SARS-CoV-2.**

| Outcome | NSAID users | | Non-users | | Comparison | | | |
|---|---|---|---|---|---|---|---|---|
| | Number of events/ sample size | Risk (%) (95% CI) | Number of events/ sample size | Risk (%) (95% CI) | Risk difference (%) (95% CI) | *p*-Value | Risk ratio (95% CI) | *p*-Value |
| **Unmatched cohort** | | | | | | | | |
| Death | 78/5,574 | 1.4 (1.1, 1.7) | 2,830/195,840 | 1.4 (1.4, 1.5) | −0.0 (−0.4, 0.3) | 0.77 | 0.97 (0.77, 1.21) | 0.078 |
| Hospitalization* | 757/5,091 | 14.9 (13.9, 15.8) | 18,543/186,479 | 9.9 (9.8, 10.1) | 4.9 (3.9, 5.9) | <0.001 | 1.50 (1.40, 1.60) | <0.001 |
| ICU admission* | 63/5,547 | 1.1 (0.9, 1.4) | 1,247/195,332 | 0.6 (0.6, 0.7) | 0.5 (0.2, 0.8) | <0.01 | 1.78 (1.38, 2.29) | <0.001 |
| Mechanical ventilation* | 38/5,558 | 0.7 (0.5, 0.9) | 698/195,582 | 0.4 (0.3, 0.4) | 0.3 (0.1, 0.5) | <0.01 | 1.92 (1.38, 2.65) | <0.001 |
| Renal replacement therapy* | 7/5,573 | 0.1 (0.0, 0.2) | 166/195,754 | 0.1 (0.1, 0.1) | 0.0 (−0.1, 0.1) | 0.39 | 1.48 (0.70, 3.15) | 0.31 |
| **Matched cohort** | | | | | | | | |
| Death | 61/5,018 | 1.2 (0.9, 1.5) | 382/20,072 | 1.9 (1.7, 2.1) | −0.7 (−1.1, −0.3) | <0.001 | 0.64 (0.49, 0.84) | <0.01 |
| Hospitalization* | 651/4,613 | 14.1 (13.1, 15.1) | 2,261/18,892 | 12.0 (11.4, 12.5) | 2.1 (1.0, 3.3) | <0.001 | 1.18 (1.08, 1.28) | <0.001 |
| ICU admission* | 50/4,992 | 1.0 (0.7, 1.3) | 144/20,024 | 0.7 (0.6, 0.9) | 0.3 (−0.0, 0.6) | 0.07 | 1.39 (1.00, 1.95) | 0.05 |
| Mechanical ventilation* | 29/5,003 | 0.6 (0.4, 0.8) | 95/20,050 | 0.5 (0.4, 0.6) | 0.1 (−0.1, 0.3) | 0.38 | 1.22 (0.79, 1.89) | 0.36 |
| Renal replacement therapy* | 6/5,017 | 0.1 (0.0, 0.2) | 16/20,060 | 0.1 (0.0, 0.1) | 0.0 (−0.1, 0.1) | 0.45 | 1.50 (0.58, 3.89) | 0.41 |

NSAID use was defined as a filled prescription within 30 days prior to the date of cohort entry.

*Patients with a secondary outcome occurring during the exclusion assessment window were excluded, resulting in exclusion of *n* = 9,844 patients for hospitalization, *n* = 535 for ICU admission, *n* = 274 for mechanical ventilation, and *n* = 87 for renal replacement therapy in unmatched cohorts, and *n* = 1,585, 74, 37, and 13, respectively, in matched cohorts.

ICU, intensive care unit; NSAID, non-steroidal anti-inflammatory drug; SARS-CoV-2, severe acute respiratory syndrome coronavirus 2.

## Strengths and limitations

The Danish nationwide registries allowed for identification of all individuals who had been tested for SARS-CoV-2 in Denmark and allowed for obtaining data on prescription drug use, medical history, migration, hospital admissions, and death through individual-level linkage between health and administrative registries. Thus, we were able to include SARS-CoV-2-positive individuals managed in the community, unlike other data sources that mainly record hospitalized patients. Further, the SARS-CoV-2 test-negative population allowed us to conduct a negative control analysis.

The main limitations of our study are potential misclassification of non-users as NSAID users; potentially unmeasured confounding due to a lack of information on study participants' bodyweight, an important risk factor of severe COVID-19; and confounding by indication due to NSAIDs possibly being prescribed due to early symptoms of severe COVID-19.

In Denmark, NSAIDs can only be obtained via prescription, except for low-dose (200 mg) ibuprofen. With the limited availability and use of NSAIDs over-the-counter in Denmark, misclassification from over-the-counter purchases of NSAIDs is of less concern [19]. However, exposure misclassification may still be present since information on adherence, intended duration, and dose was not available. Some patients do not use NSAIDs continuously after filling a

prescription, and consequently some of these patients will have been incorrectly classified as NSAID users at the time of the positive test for SARS-CoV-2. The fact that a patient had filled a prescription for an NSAID may be considered an indicator of availability of NSAIDs rather than of actual use of NSAIDs. As information on exposure status was collected prior to our outcomes of interest, the misclassification is nondifferential and may introduce a bias towards the null. This could lead to wrongly dismissing a causal detrimental effect of NSAIDs on the prognosis of COVID-19 if many NSAID users truly were non-users.

The time of cohort entry was defined by the SARS-CoV-2 test date because information on time of symptom onset was not available. Thus, the timing of NSAID use relative to cohort entry will not necessarily reflect NSAID use in the early course of COVID-19 disease.

Users of NSAIDs were more likely to be overweight or obese than non-users, both among individuals who tested positive for SARS-CoV-2 and those who tested negative. This is possibly explained by the fact that a diagnosis of overweight or obesity in the Danish National Patient Registry is dependent on a hospital admission or outpatient clinic visit. This is also a likely explanation for the relatively low prevalence of overweight or obesity in this study. The positive predictive value of these diagnoses in the registries is, however, high [28].

Use of NSAIDs has been associated with lower mortality among elderly individuals, presumably because of confounding by indication [29], i.e., NSAIDs are preferentially prescribed to younger, less frail patients because of the established renal, gastrointestinal, and cardiovascular adverse effects [30]. Considerable media attention to the hypothesized risks of use of NSAIDs in COVID-19 may also have influenced how physicians prescribed NSAIDs to selected patients. Finally, severe symptoms early in the course of COVID-19 disease, before the patient is known to be infected with SARS-CoV-2, may increase the likelihood of being prescribed NSAIDs, which would bias the effect estimates towards an increased risk of severe disease associated with NSAIDs.

The secondary outcomes reflecting in-hospital treatment decisions may be more prone to confounding by indication because of the clinical selection of patients to be hospitalized or admitted to the ICU. For example, besides disease severity, factors such as age, comorbidity, and expected outcomes are involved in the ICU triage decision.

Ideally, confounding by indication would be mitigated using an active comparator; however, a suitable active comparator does not exist for ibuprofen. In a previous study, users of paracetamol differed more from NSAID users than did non-users of NSAIDs [9].

## Findings in relation to other studies

The lack of association between use of NSAIDs and adverse outcomes in individuals who tested positive for SARS-CoV-2 in our study could be explained by multiple factors. First, NSAIDs may not increase angiotensin converting enzyme 2 in humans. The original hypothesis stems from experiments conducted in diabetic rats [31], and the findings may not transfer between organisms. Second, increased angiotensin converting enzyme 2 expression may not affect the risk of severe COVID-19. Studies on angiotensin converting enzyme inhibitors and angiotensin receptor blockers and the risk of contracting COVID-19 or experiencing a severe course of disease have not shown any association [32–34]. Third, the adverse effects of NSAIDs on the course of pneumonia may be specific to bacterial infections. A recent study on the risk of adverse outcomes in users of NSAIDs hospitalized for influenza found no association between use of NSAIDs and ICU admission or death [9], similar to the findings in this study.

Considering the available evidence, there is no reason to withdraw well-indicated use of NSAIDs during the SARS-CoV-2 pandemic. However, the well-established adverse effects of NSAIDs, particularly their renal, gastrointestinal, and cardiovascular effects, should always be

considered, and NSAIDs should be used in the lowest possible dose for the shortest possible duration for all patients [30].

## Conclusions

Use of NSAIDs was not associated with an increased risk of 30-day mortality or adverse outcomes in patients infected with SARS-CoV-2 in this cohort of all Danish residents who tested positive for SARS-CoV-2.

## Supporting information

**S1 STROBE Checklist. STROBE checklist of items that should be included in reports of cohort studies.**
(DOCX)

**S1 Appendix. Detailed information on registries used in this study.**
(DOCX)

**S2 Appendix. Codes used to define exposure, outcome, and covariate variables.**
(DOCX)

**S1 Table. Univariate association with death (RRcd) and use of NSAIDs (RRce) for each covariate in the PS.**
(DOCX)

**S2 Table. Association between current NSAID use and mortality, hospitalization, ICU admission, mechanical ventilation, and renal replacement therapy using an outcome assessment window of 60 days for mortality and 30 days for secondary outcomes.**
(DOCX)

**S3 Table. Association between use of NSAIDs within 60 days before cohort entry and 30-day mortality, hospitalization, ICU admission, mechanical ventilation, and renal replacement therapy.**
(DOCX)

**S4 Table. Association between NSAID prescription fills within the period 60 days to 14 days before cohort entry and 30-day mortality, hospitalization, ICU admission, mechanical ventilation, and renal replacement therapy.**
(DOCX)

## Acknowledgments

The departments of clinical microbiology throughout Denmark are acknowledged for contributing to the national infectious disease surveillance. The Danish Health Data Authority and Statens Serum Institut are acknowledged for valuable support with preparation and linkage of data.

## Author Contributions

**Conceptualization:** Lars Christian Lund, Kasper Bruun Kristensen, Mette Reilev, Steffen Christensen, Reimar Wernich Thomsen, Christian Fynbo Christiansen, Nanna Borup Johansen, Jesper Hallas, Anton Pottegård.

**Formal analysis:** Lars Christian Lund, Kasper Bruun Kristensen.

**Methodology:** Lars Christian Lund, Kasper Bruun Kristensen, Reimar Wernich Thomsen, Christian Fynbo Christiansen, Nanna Borup Johansen, Jesper Hallas, Anton Pottegård.

**Project administration:** Anton Pottegård.

**Supervision:** Steffen Christensen, Reimar Wernich Thomsen, Christian Fynbo Christiansen, Henrik Støvring, Nikolai Constantin Brun, Jesper Hallas, Anton Pottegård.

**Writing – original draft:** Lars Christian Lund, Kasper Bruun Kristensen, Anton Pottegård.

**Writing – review & editing:** Lars Christian Lund, Kasper Bruun Kristensen, Mette Reilev, Steffen Christensen, Reimar Wernich Thomsen, Christian Fynbo Christiansen, Henrik Støvring, Nanna Borup Johansen, Nikolai Constantin Brun, Jesper Hallas, Anton Pottegård.

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
