## [Editor Report · Decision Letter 0]

16 Jun 2020

Dear Dr Pottegaard, 

Thank you for submitting your manuscript entitled "Adverse Outcomes and Mortality in Users of Non-Steroidal Anti-Inflammatory Drugs tested positive for SARS-CoV-2: A Danish Nationwide Cohort Study" for consideration by PLOS Medicine.

Your manuscript has now been evaluated by the PLOS Medicine editorial staff and I am writing to let you know that we would like to send your submission out for external peer review.

Kind regards,

Artur Arikainen,

Associate Editor

PLOS Medicine

---

## [Decision Letter · Decision Letter 1]

30 Jun 2020

Dear Dr. Pottegaard,

Thank you very much for submitting your manuscript "Adverse Outcomes and Mortality in Users of Non-Steroidal Anti-Inflammatory Drugs tested positive for SARS-CoV-2: A Danish Nationwide Cohort Study" (PMEDICINE-D-20-02753R1) for consideration at PLOS Medicine. 

[LINK]

In light of these reviews, I am afraid that we will not be able to accept the manuscript for publication in the journal in its current form, but we would like to consider a revised version that addresses the reviewers' and editors' comments. Obviously we cannot make any decision about publication until we have seen the revised manuscript and your response, and we plan to seek re-review by one or more of the reviewers. 

We expect to receive your revised manuscript by Jul 21 2020 11:59PM. Please email us (plosmedicine@plos.org) if you have any questions or concerns.

We look forward to receiving your revised manuscript. 

Sincerely,

Emma Veitch, PhD

PLOS Medicine

On behalf of Clare Stone, PhD, Acting Chief Editor,

PLOS Medicine

plosmedicine.org

*In the last sentence of the Abstract Methods and Findings section, please include a brief note regarding any key limitation(s) of the study methods overall.

*At this stage, we ask that you include a short, non-technical Author Summary of your research to make findings accessible to a wide audience that includes both scientists and non-scientists. The Author Summary should immediately follow the Abstract in your revised manuscript. This text is subject to editorial change and should be distinct from the scientific abstract. Please see our author guidelines for more information: https://journals.plos.org/plosmedicine/s/revising-your-manuscript#loc-author-summary

*As noted by one reviewer, it would be good to include in the Discussion a more explicit and thorough discussion of the main limitations of the study methods to answer the question of interest.

*As the paper reports analyses of data from a cohort study, it would be good to ensure that the study is reported according to the STROBE guideline; if doing this please include the completed STROBE checklist as Supporting Information. Please add the following statement, or similar, to the Methods: "This study is reported as per the Strengthening the Reporting of Observational Studies in Epidemiology (STROBE) guideline (SChecklist)." The STROBE guideline can be found here: http://www.equator-network.org/reporting-guidelines/strobe/. When completing the checklist, please use section and paragraph numbers, rather than page numbers.

Comments from the reviewers:

Reviewer #1: I greatly enjoyed reading this manuscript, which is very neatly written with the entire emphasis on solid statistical evaluation, as I would expect. Some of the English is cumbersome and sloppy (eg. abstract: "matched to up to...") but on the whole it is an excellent paper, and a useful addition to the literature.

Considering the methods used and the 'solidity' of the database I have some suggestions, and I am deliberately keeping this brief:

1. It would be helpful if statins could be studied too, using almost identical methods. The title would need to be changed to reflect this.

2. In view of the RECOVERY data on dexamethasone, can steroids be looked at too? These data should at least be mentioned. As per 1.

3. It was disappointing not to see a paragraph or 2, in the introduction and discussion, on mechanisms here that may be relevant and operational, leading the authors to study this as more than a simple correlation ie. is there any suggestion that the potential association may have been causative and problematic, based on the first line of the abstract? This is not expanded upon anywhere in the text and I do not see clinicians concerned any more about NSAID use (this is an old story as the authors know).

4. There is no substantial limitations paragraph with the usual dose, duration etc discussion.

J Stebbing

Reviewer #2: I confine my remarks to statistical aspects of this paper. These were very well done and I recommend publication.

Peter Flom

Reviewer #3: This is an interesting and relevant observational study to estimate if there is any effect on patients who obtained a prescription for NSAIDs 30 days prior to a positive SARS-CoV-2 PCR test on mortality within 30 days (primary outcome) or hospitalisation, ICU admission, ventilation or acute renal replacement therapy within 14 days. This has been the subject of much debate during the current COVID-19 outbreak so represents a valuable contribution to the field and will likely impact on clinical management or public health policy.

The set of baseline covariates were defined (age, gender, comorbidities, use of other selected medications plus phase of the outbreak) and the methodology is appropriate. Propensity scores were used to ensure that the 'treated' subjects are similar in their baseline covariates to 'untreated' subjects (matched 1: 4 in this case) in order to provide an unbiased estimate of the relative risk of NSAIDs on the measured outcomes. Additional stratification looked at age (<65y; >65y), gender, cardiovascular disease and non-healthcare worker.

The analyses do not indicate that NSAIDs presented an increased risk of mortality or morbidity in the COVID-19 population in Denmark.

[LINK]

---

## [Decision Letter · Decision Letter 2]

9 Jul 2020

Dear Dr. Pottegaard,

Thank you very much for re-submitting your manuscript "Adverse Outcomes and Mortality in Users of Non-Steroidal Anti-Inflammatory Drugs tested positive for SARS-CoV-2: A Danish Nationwide Cohort Study" (PMEDICINE-D-20-02753R2) for review by PLOS Medicine.

I have discussed the paper with my colleagues and the academic editor and it was also seen again by 1 reviewer. I am pleased to say that provided the remaining editorial and production issues are dealt with we are planning to accept the paper for publication in the journal.

[LINK]

We look forward to receiving the revised manuscript by Jul 16 2020 11:59PM. 

Sincerely,

Artur Arikainen, 

Associate Editor 

PLOS Medicine

plosmedicine.org

Requests from Editors:

1. When completing the STROBE checklist, please use section and paragraph numbers, rather than page numbers.

2. The Data Availability Statement (DAS) requires revision. If the data are owned by a third party, please provide contact information for data requests (web or email address). Note that a study author cannot be the contact person for the data.

3. Abstract:

a. Line 35: please correct to: “…use of NSAIDs was…”

b. Line 48: please include basic cohort demographics (age and sex).

c. Please quantify all results with both 95% CIs and p values.

d. Line 53 (and throughout the text): please correct to: “…ICU admission…” (no hyphen)

4. Author Summary:

a. Line 71: please spell out NSAIDs.

5. Results:

a. Please quantify all results in the text and tables with both 95% CIs and p values.

b. Lines 232-233: Please give absolute numbers as well as percentages.

6. All tables and figure: Please define all abbreviations (eg. RT-PCR, IQR, SMD, COPD, ICU, NSAIDs) in the legend or footnote.

7. Lines 397-418: Please remove these sections on Funding, Competing Interests, and Author Contributions, and ensure the information is instead filled in on the submission form.

8. Lines 420-424: Please move details on ethical approval to the Methods section.

9. Re: reference 30 listed as under review, papers cannot be listed in the reference list until they have been accepted for publication or are otherwise publically accessible (for example, in a preprint archive). The information may be cited in the text as a personal communication with the author if the author provides written permission to be named. Alternatively, please provide a different appropriate reference.

-----

Comments from Reviewers:

Reviewer #1: excellent answers, lovely paper, congratulations, justin stebbing

[LINK]

---

## [Editor Report · Decision Letter 3]

3 Aug 2020

Dear Dr Pottegaard, 

On behalf of my colleagues and the academic editor, Dr. Anne C Cunningham, I am delighted to inform you that your manuscript entitled "Adverse outcomes and mortality in users of non-steroidal anti-inflammatory drugs tested positive for SARS-CoV-2: A Danish nationwide cohort study" (PMEDICINE-D-20-02753R3) has been accepted for publication in PLOS Medicine. 

PRODUCTION PROCESS

PRESS

PROFILE INFORMATION

Thank you again for submitting the manuscript to PLOS Medicine. We look forward to publishing it. 

Best wishes, 

Artur Arikainen, 

Associate Editor 

PLOS Medicine

plosmedicine.org